# Effects of Factors Influencing Scar Formation on the Scar Microbiome in Patients with Burns

**DOI:** 10.3390/ijms242115991

**Published:** 2023-11-06

**Authors:** Yeongyun Jung, Hui Song Cui, Eun Kyung Lee, So Young Joo, Cheong Hoon Seo, Yoon Soo Cho

**Affiliations:** 1Burn Institute, Hangang Sacred Heart Hospital, Hallym University College of Medicine, Seoul 07247, Republic of Korea; jyg1076@hallym.ac.kr (Y.J.); bioeast007@naver.com (H.S.C.); eunlee0617@gmail.com (E.K.L.); 2Department of Rehabilitation Medicine, Hangang Sacred Heart Hospital, Hallym University College of Medicine, Seoul 07247, Republic of Korea; anyany98@gmail.com (S.Y.J.); chseomd@gmail.com (C.H.S.)

**Keywords:** burn, burn scar, microbial community composition, factors influencing scar formation, skin microbiome

## Abstract

Skin microbiome dysbiosis has deleterious effects, and the factors influencing burn scar formation, which affects the scar microbiome composition, are unknown. Therefore, we investigated the effects of various factors influencing scar formation on the scar microbiome composition in patients with burns. We collected samples from the burn scar center and margin of 40 patients with burns, subgrouped by factors influencing scar formation. Scar microbiome composition-influencing factors were analyzed using univariate and multivariate analyses. Skin graft, hospitalization period, intensive care unit (ICU) admission, burn degree, sex, age, total body surface area burned (TBSA), time post-injury, transepidermal water loss, the erythrocyte sedimentation rate, and C-reactive protein levels were identified as factors influencing burn scar microbiome composition. Only TBSA and ICU admission were associated with significant differences in alpha diversity. Alpha diversity significantly decreased with an increase in TBSA and was significantly lower in patients admitted to the ICU than in those not admitted to the ICU. Furthermore, we identified microorganisms associated with various explanatory variables. Our cross-sectional systems biology study confirmed that various variables influence the scar microbiome composition in patients with burns, each of which is associated with various microorganisms. Therefore, these factors should be considered during the application of skin microbiota for burn scar management.

## 1. Introduction

Millions of people suffer from postsurgical skin damage, scarring from trauma, or burns annually [1]. Burn injuries cause functional and cosmetic problems by triggering abnormal wound-healing processes, often leading to raised, erythematous, and itchy hypertrophic scars [2]. These problems can have considerable physical and psychological effects and substantially reduce patients’ quality of life [3,4]. Therefore, the management of burn scars is important.

Notably, patient, injury, and treatment characteristics influence scar formation [5]. However, the pathophysiology, taxonomy, and clinical course of post-burn scarring vary [5]. Current burn scar treatments include surgical and nonsurgical methods, such as laser therapy, steroid injections, and compression therapy. However, these treatments have limited efficacy and often fail to manage symptoms [6]. The difficulty in treating morbid scarring after burns is that it is caused by various risk factors; extensive prospective studies on these risk factors are limited [7]. Consequently, long-term studies are needed to improve the outcomes of burn scar treatment and patients’ quality of life. Furthermore, new technologies are required to improve outcomes [6,8].

As the human microbiome is associated with human health and disease, considerable research has been conducted in this field in recent decades. With the advancement in the understanding of the role of microbial communities, research on disease treatment through microbiome regulation has been conducted [9]. For example, treating *Clostridium difficile* infections through fecal microbiome transplantation has achieved a cure rate of >90.000% [10]. However, treating various skin diseases using skin microbial community control is challenging [11]. Strategies for controlling skin microbial communities include skin microbiome transplantation, bacteriotherapy, and prebiotic stimulation [11]. These approaches have been used to treat acne, atopic dermatitis, and underarm odor and have achieved positive outcomes [12,13,14].

Furthermore, some studies have investigated the effect of burn injuries on the skin microbiome and have reported that burn injuries lead to the emergence of a distinct skin microbiome in patients with burns, compared with that in controls [15,16,17]. However, burns are understudied relative to other skin disorders, and the mechanism through which the skin microbiome affects patients with burns remains unknown. Factors that influence scar formation after burn injury include patient characteristics, such as sex, age, and skin type, and injury and treatment characteristics, such as burn severity, the length of hospitalization, and the type of surgery [5]. These factors also affect the skin microbiome [18,19,20,21,22]. Nevertheless, there is a need to verify which of these factors have a marked influence on the composition of the skin microbiome to gain a better understanding of the skin microbiome in patients with burns.

In the present study, we conducted a comprehensive analysis of biological systems using a substantial patient dataset to identify key clinical factors that sustantially affect the composition of the skin microbiome in patients with burns. Our findings provide a novel microbiological perspective on the formation of burn scars, based on an analysis of the skin microbiome in patients with burns.

## 2. Results

### 2.1. Study Cohort

In total, 80 burn scar samples (40 from the central areas of the scars and 40 from the margin areas of the scars) were collected from 40 patients with burns. 16S rRNA amplicon sequencing revealed that the scar microbiomes of the samples overall included species belonging to 19 phyla, 180 families, and 364 genera (Figure 1).

Of the 40 patients, 23 were male, and 17 were female patients. Skin grafting was performed in 50.000% of the patients. In addition, 25.000% of the participants had a history of intensive care unit (ICU) admission. Data were collected from patients with various burn severities: 22.500% had superficial second-degree burns, 32.500% had deep second-degree burns, and 45.000% had third-degree burns (Appendix A).

The patients were classified into subgroups to compare their characteristics in more detail (Table 1 and Appendix A). The patients with superficial second-degree burns did not undergo skin grafting, 2 of the 13 patients with deep second-degree burns underwent skin grafting, and all patients with third-degree burns underwent skin grafting. Compared with the other severity groups (patients with superficial second-degree and deep second-degree burns), the patients with third-degree burns had a significantly higher total body surface area burned (TBSA) (*p* = 0.006), longer hospitalization period in the Department of Burn Surgery (*p* = 0.004), and greater scar thickness (*p* = 0.030).

### 2.2. Differences in Microbial Composition among Groups (Beta Diversity)

A canonical correspondence analysis (CCA) was used to confirm whether patient and scar biomechanical characteristics affected the composition of the skin microbial community. The treatment methods after burn injury (skin graft, the length of hospitalization, and the length of ICU stay), burn severity (burn degree and TBSA), patient characteristics (sex and age), the degree of inflammation (erythrocyte sedimentation rate and C-reactive protein level), and time after burn injury were found to affect the beta diversity of the skin microbiome composition significantly (Figure 2).

These results suggest that the factors influencing scar formation (patient, injury, and treatment characteristics) also influence the scar microbiome composition. Except for transepidermal water loss (TEWL), the other scar biomechanical characteristics did not yield significant results. Skin hydration, melanin, itch numerical rating scale (NRS), erythema, and thickness did not significantly affect the scar microbial community. In addition, no significant differences were observed in the microbiome between the scar center and margin.

### 2.3. Differences in Scar Microbial Diversity (Alpha Diversity)

Differences in species diversity (alpha diversity) were analyzed in relation to 11 variables that significantly affected the beta diversity of the skin microbial composition. Continuous variables, such as age, burn degree, TBSA, the erythrocyte sedimentation rate (ESR), the C-reactive protein (CRP) level, the length of hospitalization, time after burn injury, and TEWL, were grouped into quartiles. Regarding the duration of ICU admission, instead of quartile grouping, the patients were categorically assigned to groups according to whether they were admitted to the ICU. Nine of the eleven variables yielded no significant differences in phylogenetic diversity (Appendix A). Alpha diversity differed significantly according to ICU admission status and TBSA (Figure 3). Phylogenetic diversity was significantly (*p* = 0.005) lower in the patients admitted to ICU than in those not admitted to ICU (Figure 3b). For TBSA, Q1 (lowest TBSA) and Q4 (highest TBSA) did not show any significant difference (*p* = 0.122). However, when comparing Q2 and Q4 (*p* = 0.041), and Q3 and Q4 (*p* = 0.027), we found that the higher the TBSA, the lower the phylogenetic diversity (Figure 3d). These results indicate that TBSA and admission or non-admission to the ICU could affect the composition and diversity of the scar microbial community.

### 2.4. Relative Abundance of Categorical Variables, as Determined Using Linear Discriminant Analysis Effect Size

Linear discriminant analysis effect size (LEfSe) was used to investigate the taxonomic differences in the scar microbiome for categorical variables (LEfSe analysis, *p* < 0.01, LDA score > 3). According to the LEfSe, *Propionibacterium* was relatively more abundant in males, whereas *Acinetobacter*, *Lactobacillus*, *Bacillus*, unclassified Bacilaceae, *Pseudomonas*, unclassified Enterobacteriaceae, unclassified Bartonellaceae, and *Rhodobacter* were relatively more abundant in females (Figure 4a). The patients who underwent skin grafting had species of relatively more genera (*Enhydrobacter*, *Anaerococcus*, *Actinomyces*, and *Bdellovibrio*) than those who did not, and the patients who did not undergo skin grafting were confirmed to have more unclassified Actinomycetales (Figure 4b). The patients admitted to ICU had a higher abundance of *Dermabacter* than those who were not admitted to ICU. However, the patients admitted to ICU had a lower abundance of *Paracoccus*, *Chryseobacterium*, unclassified Comamonadaceae, unclassified Gemellaceae, *Abiotrophia*, *Pantcea*, *Janthinobacterium*, *Arthobacter*, and unclassified Pseudomonadaceae (Figure 4c). No characteristic genera were identified in relation to burn degree.

### 2.5. Identification of Continuous Variables and Associated Microbes

A regression analysis was used to determine the relationship between scar microbial communities and continuous variables (the length of hospitalization, age, TBSA, time after injury, TEWL, ESR, and CRP levels) (Figure 5). The regression analysis of the top 30 bacterial genera and continuous variables showed no significant correlations with the length of hospitalization, age, or TEWL. TBSA positively correlated with *Methylobacterium* (*R* = 0.291, *p* = 0.009) and negatively correlated with *Paracoccus* (*R* = −0.292, *p* = 0.009). Time after injury positively correlated with *Acinetobacter* (*R* = 0.370, *p* = 0.001) and *Kocuria* (*R* = 0.351, *p* = 0.001). ESR positively correlated with *Brevundimonas* (*R* = 0.369, *p* = 0.001), *Chryseobacterium* (*R* = 0.328, *p* = 0.003), and *Facklamia* (*R* = 0.331, *p* = 0.003) and negatively correlated with *Propionibacterium* (*R* = −0.309, *p* = 0.005). The CRP level positively correlated with *Chryseobacterium* (*R* = 0.490, *p* < 0.001). These results suggest that TBSA, the duration of burn injury, ESR, and CRP levels affect skin microbial communities and increase or decrease the abundance of specific skin microbes.

## 3. Discussion

The survival rate of patients with burns has improved markedly over the past few decades; however, post-burn pathological scarring remains one of the biggest challenges in the management of patients with burns [23]. Pathological scar formation after burns depends on several variables, including patient, injury, and treatment characteristics [5]. The skin microbiome exerts a wide range of effects on the immune system, barrier function, and wound-healing response of the skin, and it has anti-aging and anti-inflammatory effects [24,25,26]. Burn injuries alter the skin microbiome, affecting these processes [15]. Therefore, as the first step from association to causation, the identification of the effects of the various related factors is important. In the present study, we aimed to determine the effects of various factors influencing scar formation on the scar microbiome composition in patients with burns. We performed a systems biology analysis of a well-characterized cohort of patients with burns. The CCA analysis showed that patient, injury, and treatment characteristics are important variables affecting the scar microbial community. The effects of specific factors on community diversity and composition were also identified.

Over the past few years, several studies on the effects of burn injuries on the microbiome have been conducted [27]. Burn injuries reduce gut microbial diversity and increase intestinal permeability [28,29]. Furthermore, burn injuries increase the number of potentially pathogenic bacteria, leading to intestinal imbalance [28,29]. Skin microbiome studies have also been conducted [15,16,17]. Burn injuries are reportedly associated with a microbial community that is distinct from that of healthy skin [15,16,17]. However, conflicting results have been reported on skin microbiome diversity. According to Liu et al., more operational taxonomic units were observed in patients with burns than in controls. Furthermore, Shannon’s evenness index was higher in patients with burns than in controls [16]. Mouse experiments conducted by Sanjar et al. showed that controls had a higher alpha diversity (Chao 1, Shannon, Simpson, observed species, and phylogenetic diversity indices) than patients with burned skin [17]. These inconsistent results could be attributed to the differences between humans and mice, and such contrasting results may be related to the lack of large-scale studies. Various risk factors for burn scar formation are known; however, to date, no studies have been conducted on the correlation between these factors and the scar microbiome. Therefore, further research is required to gain a better understanding of burn injuries and the skin microbiome.

Burns are often classified as major or minor, according to TBSA. Major burns cause serious problems in the local burn area, as well as in the whole body, because of immune and inflammatory reactions and metabolic shock [8]. They are also known to alter the human microbiome [8]. The present study’s results indicate that TBSA could affect the alpha diversity of the scar microbiome. According to a previous study involving older patients with burns, those with a higher TBSA had a reduced gut microbiome diversity than those with a lower TBSA [30]. No reports on the diversity of TBSA and burn scar (skin) microbial communities have been published. Consistent with the results of previous studies on the gut microbiome, the present study’s results show that the alpha diversity of the scar microbiome decreased with a higher TBSA. Taken together, these results suggest that severe burn injuries lower microbial community diversity in the gut and skin, leading to dysbiotic conditions. *Anaerococcus* was more abundant in patients who underwent skin grafting. *Anaerococcus* spp. are common skin commensal bacteria; however, these are present in chronic skin diseases and injuries [31,32]. Thus, our finding was plausible, because patients who underwent skin grafting had a higher TBSA and a longer hospitalization period after injury than those who did not undergo skin grafting.

ICU admission affects the alpha diversity of the scar microbiome. Patients admitted to ICUs have a lower skin alpha diversity than those who are not [33,34,35]. The loss of diversity in ICU inpatients may be associated with differences in treatment modalities, such as systemic antibiotic therapy, and long-term exposure to hospital pathogens [35]. In our study, *Paracoccus* was more abundant in the patients not admitted to the ICU (Figure 4c) and negatively correlated with TBSA (Figure 4). *Paracoccus* is less prevalent in chronic wounds than in healthy skin [36]. The patients admitted to the ICU had a significantly higher TBSA than those who were not (Appendix A). These results suggest that *Paracoccus* plays an important role in shaping normal skin microbial communities.

The scar microbiome exhibits sex-specific differences in composition and alpha diversity [22]. In the present study, the phylogenetic diversity tended to be higher in females than in males, although the differences did not reach statistical significance (Appendix A). We also confirmed that the scar microbial community exhibited a lower *Acinetobacter* abundance and a higher *Propionibacterium* abundance in men than in women (Figure 4a). According to a study by Ying et al., *Acinetobacter* was significantly less prevalent in males than in females, whereas the relative abundance of *Propionibacterium* was significantly higher in males than in females [37], consistent with the findings of the present study. *Propionibacterium* is a representative commensal bacterium of the skin microbiome and is thought to confer health benefits through short-chain fatty acid production [38]. Lipophilic bacteria, such as *Propionibacterium,* prefer environments containing abundant moisture and sebum [37]. Women have lower sebum levels with increasing age, whereas men have better sebum maintenance with increasing age [39]. The participants in the present study were middle-aged adults (≥40 years) (Appendix A). Therefore, *Propionibacterium* was expected to be more abundant in males than in females.

The present study has several strengths. First, it included comprehensive patient characterization data, allowing for detailed analyses of different variables. Second, it included detailed information on patient, injury, and treatment characteristics according to each subgroup. Finally, no previous study has investigated the effects of various factors affecting scar formation on the scar microbial communities in patients with burns.

The present study also has some limitations, such as the small sample size of the cross-sectional design and subgroup analysis, rendering it difficult to conclude causality. Another limitation of our study arises from the recruitment of patients with scars of a relatively low thickness. Therefore, a second replication cohort with a much larger number of samples, more refined groupings, and sampling points is required. Furthermore, a more in-depth investigation is warranted through comparative studies involving patients exhibiting significant variations in specific biomechanical characteristics, including thickness. This would allow us to better understand how the skin microbiome influences biomechanical scar characteristics. Nevertheless, the present study can facilitate the remediation of the scar microbial community in burn scar management.

## 4. Materials and Methods

### 4.1. Study Participants and Sample Collection

This study involved an analysis of 16S rRNA amplicon sequencing data obtained from the burn scars of patients who underwent rehabilitation therapy for burns at the Department of Rehabilitation Medicine between September 2021 and July 2022. The study design and protocol were approved by the Institutional Review Board of Hangang Sacred Heart Hospital (HG2020-007) and registered in the Clinical Research Information Service registry (KCT0005228). Written informed consent for participation and publication was obtained from all participants, and all methods were performed in accordance with relevant guidelines and regulations.

Ninety-eight patients undergoing rehabilitation therapy in the Department of Rehabilitation Medicine, who had received wound dressing and surgical treatment in the Department of Burn Surgery at the burn center, were enrolled in this study. Of the 98 patients, 81 with burn scars surrounded by normal skin were selected. Then, 37 patients were excluded based on the following exclusion criteria, and 44 patients were ultimately swabbed for skin samples. The exclusion criteria were (1) the presence of infected scars (accompanied by pus) (*n* = 6), (2) the presence of electrical burns (*n* = 5), (3) the presence of metabolic diseases (diabetes, hypertension, and others) (*n* = 15), (4) the presence of dermatitis or psoriasis (*n* = 2), and (5) a history of oral and topical antibiotic treatment within 4 weeks before the study (*n* = 9). Of the 44 patients, 4 were excluded from the quality control for extracted metagenomic DNA, and a total of 80 samples (40 from the central areas of the scars and 40 from the margin of the scars) from the final 40 patients were used for amplicon sequencing.

All patients used the Korean Medical Device moisturizers for burn patients twice or three times a day as previously recommended scar management following complete re-epithelialization of the burn wound. All patients avoided bathing in the morning before swabbing. The skin samples analyzed in this study were collected from the central and margin areas of burn scars using saline-soaked sterile cotton swabs, and they were stored at −80 °C until DNA extraction. The biomechanical properties of the scars were investigated using previously described equipment [40]. Blood samples were collected after fasting for 8 h. The following characteristics were evaluated as possible microbiome-influencing factors: age, burn degree, skin graft, sex, TBSA, ESR, CRP level, the length of hospitalization in the Department of Burn Surgery, the length of stay in the ICU, admission or non-admission to the ICU, time after injury, scar type, itch NRS, scar thickness, melanin, erythema, TEWL, and skin hydration (Appendix A).

### 4.2. Total DNA Isolation, 16S Library Preparation, Sequencing, and Analysis

Total DNA was extracted from each swab head using a DNeasy PowerSoil Pro Kit (Qiagen, Hilden, Germany). The V4–V5 hypervariable region of the 16S rRNA gene was amplified using polymerase chain reaction (PCR) with the following primers (5′–3′): 515F-CGCTCTTCCGATCTGTGNCAGCMGCCGCGGTRA; 907R-GTGCTCTTCCGATCCGYCWATTYHTTTRAGTTT). Illumina sequence adapters were ligated to PCR products using a Nextera^®^XT index kit (Illumina, San Diego, CA, USA), according to the manufacturer’s protocol. The PCR products were purified using an AMPure XP bead purification kit (Beckman Coulter, Brea, CA, USA), and the concentrations of the purified PCR products were standardized. An Agilent 2100 Bioanalyzer (Agilent Technologies, Santa Clara, CA, USA) was used to determine the exact concentrations required for sequencing. All samples underwent sequencing on an Illumina MiSeq platform (Illumina) at the KNU NGS Core Facility, Kyungpook National University, Daegu, South Korea, using a MiSeq Reagent Kit v3 (300 bp paired-end reads; Illumina).

### 4.3. Statistical Analyses

For a microbiome analysis, QIIME2 [41] was used. Primer and adapter sequences were removed using the q2-cutadapt plugin [42]. The q2-quality-filter plugin was used to control sequence quality, and the q2-deblur plugin was used for denoising [43,44]. Taxonomies were assigned to representative sequences in the Greengenes database (version 13.8) using a q2-feature-classifier [45]. Rooted and unrooted phylogenetic trees were generated using Mafft, mask, and FastTree protocols [46,47]. The generated phylogenetic tree was used for a diversity analysis. The smallest sample had 10,076 features, which were subjected to α- and β-diversity analyses. Alpha diversity was assessed by calculating Shannon’s and Faith’s phylogenetic diversity indices. The variability of each variable was measured using Mann–Whitney test (sex, skin graft, and ICU admission) or Kruskal–Wallis test (age, burn degree, TBSA, ESR, CRP, the length of hospitalization, time after burn injury, and TEWL). Variables, such as age, burn degree, TBSA, ESR, CRP level, the length of hospitalization, time after burn injury, and TEWL, were grouped into quartiles. Analyses were performed using Prism 8 software (GraphPad Software, San Diego, CA, USA). Statistical significance was set at *p* < 0.05. Beta diversity was analyzed using CCA based on Bray–Curtis dissimilarity. LEfSe was used to select the genera of burn scars associated with categorical variables [48]. A Pearson correlation analysis was performed using the “Hmisc” R package to assess the correlations of continuous variables (age, TBSA, ESR, CRP, the length of hospitalization, time after burn injury, and TEWL) with the burn scar microbiome [49]. Figures were created in R using “ggplot2” and “ggpubr” [50,51]. Sequencing data are publicly available in the NCBI Sequence Read Archive (accession number: PRJNA973215).

## 5. Conclusions

In this study, we determined the effects of various factors influencing scar formation on the scar microbial community in patients with burns using a 16s rRNA analysis and next-generation sequencing technology. This cross-sectional systems biology study found that TBSA and ICU admission were major factors associated with alpha diversity changes and variations in microbial community composition. Sex, skin graft, and inflammation levels (ESR and CRP levels) were also explanatory variables associated with microbes. To date, no study has evaluated the effects of factors influencing scar formation on the scar microbiome composition with sufficient data on patient, injury, and treatment characteristics. This study revealed the importance of different factors affecting burn injuries in relation to the skin microbiome.

## Figures and Tables

**Figure 1 ijms-24-15991-f001:**
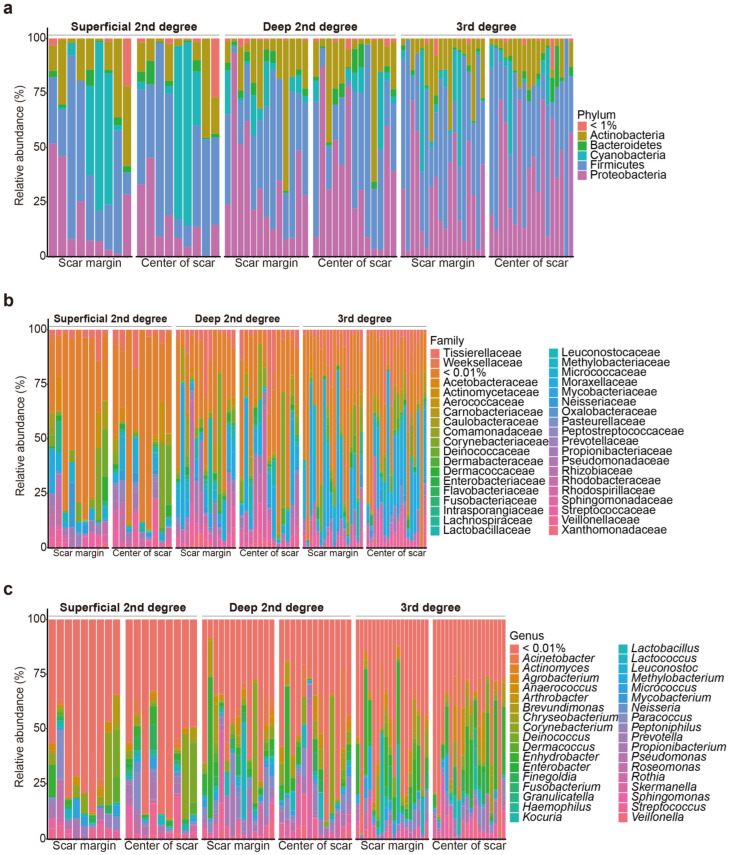
Relative microbial composition, at the phylum (**a**), family (**b**), and genus (**c**) levels, at the burn scar center and burn margins in patients with burns.

**Figure 2 ijms-24-15991-f002:**
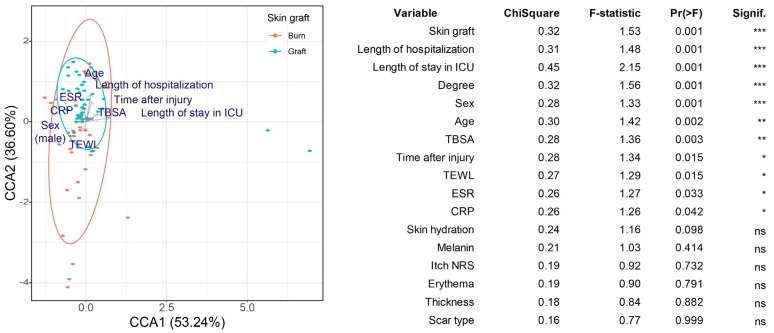
Canonical correspondence analysis (CCA) based on Bray–Curtis dissimilarity. Skin grafting was used as the grouping variable, as it yielded the lowest *p*-value in the univariate analysis. The effect of the other explanatory variables was also included in the model. The table shows the results of the CCA for variables with significant effects identified in the univariate analysis. Arrows indicate the relationship between variables. * 0.01 < *p* < 0.05, ** 0.001 < *p* ≤ 0.01, *** *p* ≤ 0.001. ns indicates *p* ≥ 0.05.

**Figure 3 ijms-24-15991-f003:**
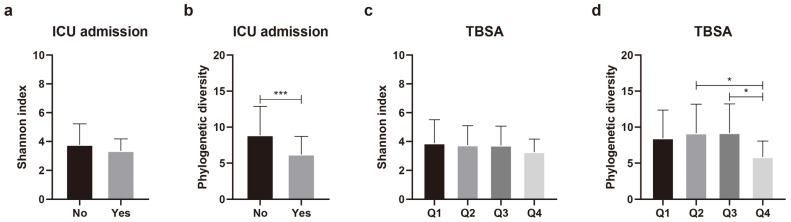
Comparison of alpha diversity of skin microbiota according to intensive care unit (ICU) admission and total body surface area (TBSA). (**a**) Shannon index and (**b**) phylogenetic diversity according to ICU admission. (**c**) Shannon index and (**d**) phylogenetic diversity according to TBSA. Values are expressed as means ± standard deviation. Asterisk (* and ***) indicates statistically significant (0.01 < *p* < 0.05 and *p* ≤ 0.001) differences between groups based on Welch’s *t*-test and Kruskal–Wallis test.

**Figure 4 ijms-24-15991-f004:**
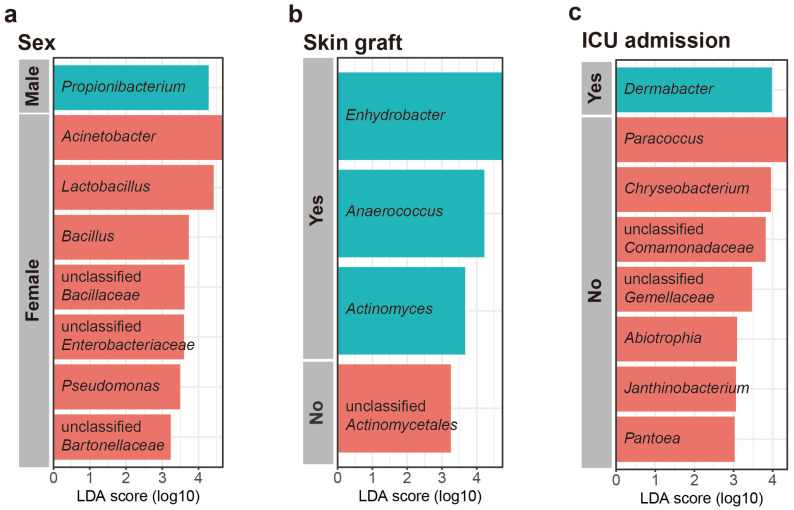
Most differentially abundant taxa selected using linear discriminant analysis effect size (LEfSe) for (**a**) sex, (**b**) skin graft, and (**c**) intensive care unit (ICU) admission.

**Figure 5 ijms-24-15991-f005:**
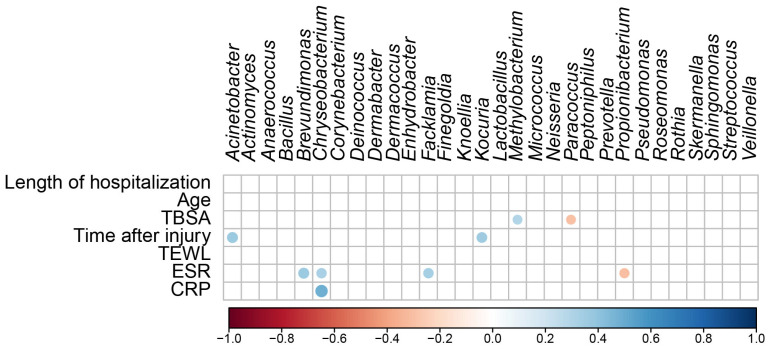
Correlation matrix shows significant correlations (*p* < 0.05) between the different microbial taxa in the differential abundance analysis and the continuous variables. The color code and size of the circles are *ρ* correlation coefficients.

**Table 1 ijms-24-15991-t001:** Biomechanical characteristics of participants and scars according to burn severity.

Variable		Burn Degree Group (Patients with Burns)			
		Superficial 2nd Degree (*n* = 9)	Deep 2nd Degree (*n* = 13)	3rd Degree (*n* = 18)	*p*
**Age, years**		41.56 ± 13.69	46.69 ± 11.79	48.06 ± 10.93	0.488
**Sex**	Male	5 (55.56%)	8 (61.54%)	10 (55.56%)	0.938
	Female	4 (44.44%)	5 (38.46%)	8 (44.44%)	
**Skin graft**	Yes	0 (0.00%)	2 (15.38%)	18 (100.00%)	<0.001
	No	9 (100.00%)	11 (84.62%)	0 (0.00%)	
**TBSA burned, %**		10.67 ± 8.72	12.00 ± 9.44	27.33 ± 16.87	0.006
**ESR, mm/H**		10.67 ± 8.05	10.08 ± 7.94	13.00 ± 8.34	0.563
**CRP, mg/L**		0.93 ± 0.81	2.16 ± 2.93	2.92 ± 3.79	0.312
**Length of hospitalization, days**		15.44 ± 11.56	18.46 ± 12.57	38.67 ± 20.35	0.004
**ICU admission**	Yes	2 (22.22%)	1 (7.69%)	7 (38.89%)	0.138
	No	7 (77.78%)	12 (92.31%)	11 (61.11%)	
**Time after injury, days**		73.75 ± 51.29	69.46 ± 37.66	137.40 ± 74.36	0.006
**Variables**		**Burn Degree Group (Scar Samples)**			
		**Superficial 2nd Degree (*n* = 18)**	**Deep 2nd Degree (*n* = 26)**	**3rd Degree (*n* = 36)**	** *p* **
**Scar type**	Center of scar	9 (50.00%)	13 (50.00%)	18 (50.00%)	>0.999
	Scar margin	9 (50.00%)	13 (50.00%)	18 (50.00%)	
**Itch NRS**		1.67 ± 2.47	1.85 ± 2.46	2.86 ± 3.37	0.453
**Thickness, mm**		0.02 ± 0.03	0.07 ± 0.12	0.18 ± 0.20	0.030
**Melanin, AU**		145.20 ± 56.39	176.80 ± 79.28	148.00 ± 61.60	0.352
**Erythema, AU**		258.70 ± 131.20	325.30 ± 148.90	305.30 ± 143.10	0.305
**TEWL, g/m^2^h**		10.99 ± 3.67	11.60 ± 3.67	12.29 ± 4.55	0.620
**Skin hydration, AU**		48.73 ± 14.82	44.60 ± 13.58	45.59 ± 16.33	0.772

Data are expressed as mean  ±  standard deviation or *n* (percentage). The *p*-value was obtained using Welch’s *t*-test and the chi-square test for continuous and categorical variables, respectively. Statistical significance was set at *p* < 0.05. Abbreviations: TBSA, total body surface area; ESR, erythrocyte sedimentation rate; CRP, C-reactive protein; ICU, intensive care unit; NRS, numerical rating scale; TEWL, transepidermal water loss; AU, arbitrary unit.

## Data Availability

The datasets generated and/or analyzed during the current study are available on the NCBI Sequence Read Archive database (accession number: PRJNA973215).

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
