# Peer review of "Effects of Factors Influencing Scar Formation on the Scar Microbiome in Patients with Burns"

_ijms, 2023, doi:10.3390/ijms242115991_

Round 1
Reviewer 1 Report
Comments and Suggestions for Authors
The authors have investigated a "newer" concept relating microbiomes to health: in this case the surface microbiome of post-burn scar. I was interested in the details of the methodology which were not forthcoming: how did they select patients (they mention their exclusion criteria); how were the areas of interest to be surface swabbed selected? They mention scar centres and periphery: is periphery mean margins of the scar or unwounded normal skin? Did they wash the area before swabbing? what moisturisers had been used for the healed wounds/scars and how long prior to sampling had the scars been moisturised; did that make any difference given topical moisturisers are a norm for post-burn scar management? Further more, they have sampled blood at the time of surface swabbing for microbiology: why did they detect a variance in ESR or CRP? were the patients not fully healed at the time of the study? Did they have any other pathology that was contributing to the variance in ESR/CRP?
They mention the thickness of the scar: presume that is the area of interest that they have sampled: thickness does not seem to be more than 2mm in height? how did they classify hypertrophic scarring? Further more, the time after burn injury has been logged but NOT time after complete healing which dictates transition from wound to scarring. In other words were all scars of the same age range?
It is interesting to read that there are differences in the microbiome of the patient if they have had ICU stay or not and if they have had skin grafts or not which to a certain extent confirms a long held view amongst clinicians. In their discussion section, they have not discussed how they would speculate the alteration of the microbiome might influence the development of type of scar or indeed the time to maturation of the scar. Whilst I acknowledge these studies are time consuming and expensive, longitudinal alterations of the microbiome in a burn patient would be paramount to understanding the evolution of the effect on change.
Comments on the Quality of English Languagegenerally good: some minor errors e.g: line 35: should be raised not swollen; lines 42-45: unsure what the meaning of the statement is; repetition of sentences particularly in the discussion section.
Author Response
Notes for the paper corrected according to the reviewers
Dear reviewers,
We sincerely thank you for your great efforts in reviewing our paper. According to your and the editor's comments, the paper has been thoroughly corrected. Our detailed point-by-point responses are provided below. Our revisions to the manuscript have been marked using yellow highlights.
Comments for reviewer #1
The authors have investigated a “newer” concept relating microbiomes to health: in this case the surface microbiome of post-burn scar. I was interested in the details of the methodology which were not forthcoming:
Response: We sincerely appreciate your great comment. We revised the methodology in detail following your comments.
how did they select patients (they mention their exclusion criteria)
Response: Thanks for your comment. Ninety-eight patients undergoing rehabilitation therapy in the Department of Rehabilitation Medicine, who had received wound dressing and surgical treatment in the Department of Burn Surgery at the burn center, were enrolled in this study. Of the 98 patients, 81 with burn scars surrounded by normal skin were selected. 37 patients were excluded based on exclusion criteria, and 44 patients were ultimately swabbed for skin samples. Of the 44 patients, 4 were excluded from the Quality Control for extracted metagenomic DNA, and a total of 80 samples from the final 40 patients were used for amplicon sequencing. We have added this to the Materials and Methods of this manuscript (pages 9-10, lines 288-300).
how were the areas of interest to be surface swabbed selected? They mention scar centres and periphery: is periphery mean margins of the scar or unwounded normal skin?
Response: We appreciate your comment. Changes in the microbial community of unwounded normal skin have been reported in burn patients (reference 15). As the margin area of the scar may be influenced by the surrounding normal skin microbiota, we collected the skin samples on the central and margin areas of burn scars, respectively. Therefore, periphery means the margin of the scar. We have added this to the Materials and Methods of this manuscript (page 10, lines 304-305).
Did they wash the area before swabbing? what moisturisers had been used for the healed wounds/scars and how long prior to sampling had the scars been moisturised; did that make any difference given topical moisturisers are a norm for post-burn scar management?
Response: Thanks for the good points. All patients avoided bathing in the morning before swabbing according to the previous studies. All patients were instructed to use the Korean Medical Device moisturizer for burn patients twice or three times a day as previously recommended scar management following complete re-epithelialization of the burn wound. The Korean Medical Device moisturizers, certified by the Ministry of Health and Welfare, can help to repair the skin barrier and microbial composition damaged from burn injury. We have revised the Materials and Methods of this manuscript (page 10, lines 301-304).
Furthermore, they have sampled blood at the time of surface swabbing for microbiology: why did they detect a variance in ESR or CRP? were the patients not fully healed at the time of the study? Did they have any other pathology that was contributing to the variance in ESR/CRP?
Response: Thanks for your comment. Many factors associated with burn injury can cause pathological scarring in burn patients. The prolonged inflammation can induce cytokine and growth factors release and it is a common mechanism of hypertrophic scarring postburn. Therefore, we evaluated a variance in ESR or CRP, which was not a significant difference among burn degree groups (p<0.05). All participants did not have any other pathology.
Reference: Zhu Z, et al. The molecular basis of hypertrophic scars. Burns Trauma 2016;4:2. Gauglitz GG, et al. Hypertrophic scarring and keloids: pathomechanisms and current and emerging treatment strategies. Mol Med 2011;17:113–25.
They mention the thickness of the scar: presume that is the area of interest that they have sampled: thickness does not seem to be more than 2mm in height? how did they classify hypertrophic scarring?
Response: Thanks for the good point. We aimed to evaluate if the factors, alread known to influence the scar formation after burn injury, can affect the composition of the scar microbiome. Thus, various degrees of burn injury from superficial 2nd degree to 3rd degree was initially included in this study, and as a result, scars with less than 2mm of height were also included. We are working on a follow-up paper observing hypertrophic changes in scar formation over time. According to VanCouver Scar Scale (VSS), the scars in the follow-up paper are classified as flat, < 2mm, 2~5 mm, and >5 mm. Because this study has the limitation that many thin-thickness scars were included, we changed the title of “Effects of factors influencing hypertrophic scar formation on the scar microbiome of patients with burns” to the title of “Effects of factors influencing scar formation on the scar microbiome of patients with burns” (page 1, lines 2-3). Additionally, we changed “factors influencing hypertrophic scar formation” to “factors influencing scar formation” (pages 1-11). Thanks for your comment again.
Furthermore, the time after burn injury has been logged but NOT time after complete healing which dictates transition from wound to scarring. In other words, were all scars of the same age range?
Response: We highly appreciate your comment. As you said, the time after burn injury means the time from burn injury to obtain the skin samples but does not mean the time from burn injury to complete healing of the burn wound. All scars in this study were of different age range. As burn patients had different severity of burn injury, the time from burn injury to complete healing was different between all patients. To reduce misunderstanding, we deleted the sentence of “All recruited participants had complete re-epithelialization of burn wounds, as confirmed by two experts” (page 9, lines 288). Additionally, we revised the part of the study participants and sample collection in the Materials and Methods of this manuscript (pages 9-10, lines 288-305).
It is interesting to read that there are differences in the microbiome of the patient if they have had ICU stay or not and if they have had skin grafts or not which to a certain extent confirms a long held view amongst clinicians. In their discussion section, they have not discussed how they would speculate the alteration of the microbiome might influence the development of type of scar or indeed the time to maturation of the scar. Whilst I acknowledge these studies are time consuming and expensive, longitudinal alterations of the microbiome in a burn patient would be paramount to understanding the evolution of the effect on change.
Response: Thanks for the good point. We selected patients with burn scars surrounded by normal skin to obtain scar margin samples and as a result, patients with relatively low height of scar thickness were recruited as subjects. As you said, this presented a limitation in finding out how changes in the microbiome composition of the burn scar affected the pathological scarring or, indeed, the time to scar maturation. Therefore, we are trying to recruit patients with higher scar thickness in future studies to see changes in the microbial community and scar biomechanical properties over time. We have added this limitation to the Discussion of this manuscript (page 9, lines 269 – 275). Thanks again for the good point.
Reviewer 2 Report
Comments and Suggestions for Authors
Dear Author,
It is with great pleasure that I review the manuscript. However, when I read the title, I was very interested to find out:
1. How the microbiome is different in patients with burn skin vs normal skin. 2. How the difference in microbiome affects the scar formation.
Please elaborate on two of these questions
Thanks
Author Response
Notes for the paper corrected according to the reviewers
Dear reviewers,
We sincerely thank you for your great efforts in reviewing our paper. According to your and the editor's comments, the paper has been thoroughly corrected. Our detailed point-by-point responses are provided below. Our revisions to the manuscript have been marked using yellow highlights.
Comments for reviewer #2
It is with great pleasure that I review the manuscript.
Response: Thank you for your kind comments.
However, when I read the title, I was very interested to find out:
- How the microbiome is different in patients with burn skin vs normal skin.
Response: Burn injuries are known to cause dysbiosis in the skin microbiome. It is known that alpha diversity does not change significantly, but beta diversity changes due to changes in microbial composition. Staphylococcus sp. is known to overgrowth in burn scars compared to normal skin. However, these studies have relatively small numbers of subjects and do not control for various burn injury factors, so additional research is needed.
Reference: Liu, Su‐Hsun, et al. "The skin microbiome of wound scars and unaffected skin in patients with moderate to severe burns in the subacute phase." Wound Repair and Regeneration 26.2 (2018): 182-191. Yu, Jiarong, et al. "Microbiome dysbiosis occurred in hypertrophic scars is dominated by S. aureus colonization." Frontiers in Immunology 14 (2023).
- How the difference in microbiome affects the scar formation.
Response: It is not yet clear how the skin microbiome influences scar formation. Additionally, no studies have yet investigated the impact of the skin microbiome on scar formation. We are therefore working on a follow-up paper analyzing each patient's initial skin microbiome and observing changes in scar formation over time.
Reviewer 3 Report
Comments and Suggestions for Authors
The manuscript describes an approach to understanding the effects of factors influencing hypertrophic scar formation on the scar microbiome of patients with burns. The authors' findings reveal that various variables influence the scar microbiome composition in patients with burns, each of which is associated with various microorganisms. Therefore, these factors should be considered during the application of skin microbiota for burn treatments
The study is important, well-designed, and strong concerning the matter in discussion.
Some minor changes should be made in order to publish the manuscript:
1) Page 1 Line 32: The authors are going to speak about other traumas why start with burns? Other traumas and then burns again? It would be better to start with the general and then proceed to the particular.
2) Page 2, Line 48: Where it is written “conducted in the field in recent decades” it should be “conducted in this field in recent decades”.
3) Page 2, Line 58: Where it is written “formation of a differentiated skin microbiome in patients with burns.”, the authors should altered the “differentiated” word once that it means that the cells had differentiated which is not the case once they are microorganisms.
4) Page 2, Line 78-82: Please always place the percentages with the same number of significant figures.
5) Page 2, Line 87: Where it is written: “Compared with other severity groups, patients.” Please explain what in more detail.
6) Page 2, Line 88: Explain the acronym the first time it is used (like in TBSA it is explained in table 1 after the first time). This problem is consistent throughout the text.
7) Page 4, Line 103: Idem to comment 6. Explain the acronym the first time it is used (like in CCA it is explained in Figure 2 caption after the first time). This problem is consistent throughout the text, therefore please verify all of them.
8) Page 8, Line 205: I do not understand the meaning of “Chao 1, Shannon, Simpson, observed species, and phylogenetic diversity indexes.”
After these changes the work is publishable.
Author Response
Notes for the paper corrected according to the reviewers
Dear reviewers,
We sincerely thank you for your great efforts in reviewing our paper. According to your and the editor's comments, the paper has been thoroughly corrected. Our detailed point-by-point responses are provided below. Our revisions to the manuscript have been marked using yellow highlights.
Comments for reviewer #3
The manuscript describes an approach to understanding the effects of factors influencing hypertrophic scar formation on the scar microbiome of patients with burns. The authors' findings reveal that various variables influence the scar microbiome composition in patients with burns, each of which is associated with various microorganisms. Therefore, these factors should be considered during the application of skin microbiota for burn treatments.
The study is important, well-designed, and strong concerning the matter in discussion.
Response: Thank you for your kind comments.
Some minor changes should be made in order to publish the manuscript:
- Page 1 Line 32: The authors are going to speak about other traumas. Why start with burns? Other traumas and then burns again? It would be better to start with the general and then proceed to the particular.
Response: Thanks for the good point. We wanted to write about how burns affect millions of people every year. I did not expect that starting with skin trauma would interfere with readers' understanding. Therefore, the first sentence starting with Skin trauma has been deleted (page 1, line 32).
- Page 2, Line 48: Where it is written “conducted in the field in recent decades” it should be “conducted in this field in recent decades”.
Response: Thank you for this comment. We have corrected that sentence (page 2, line 48).
- Page 2, Line 58: Where it is written “formation of a differentiated skin microbiome in patients with burns.”, the authors should alter the “differentiated” word once that it means that the cells had differentiated, which is not the case once they are microorganisms.
Response: We appreciate your comment. To reduce misunderstanding, we changed this paragraph to "the change of a distinct skin microbiome in patients with burns" (page 2, line 58).
- Page 2, Line 78-82: Please always place the percentages with the same number of significant figures
Response: Thanks for the good point. We also changed the % to be the same as the significant figures (3 decimal places) (page 2, lines 80-83).
- Page 2, Line 87: Where it is written: “Compared with other severity groups, patients.” Please explain what in more detail.
Response: Thank you for this comment. We further described it as “severity groups (patients with superficial 2nd degree and deep 2nd degree burns)” (page 2, lines 88-89).
- Page 2, Line 88: Explain the acronym the first time it is used (like in TBSA it is explained in table 1 after the first time). This problem is consistent throughout the text.
Response: We highly appreciate your comment. It's our mistake. Thanks to this, we marked all abbreviations including TBSA, CCA, and so on (page 2, lines 80-81 and 89-90; page 4, line 106; page 5, lines 121-122 and 129-130; page 6, line 150; page 10, lines 309-311).
- Page 4, Line 103: Idem to comment 6. Explain the acronym the first time it is used (like in CCA it is explained in Figure 2 caption after the first time). This problem is consistent throughout the text, therefore please verify all of them.
Response: Thank you again for your kind comments. We marked all abbreviations including TBSA, CCA, and so on (page 2, lines 80-81 and 89-90; page 4, line 106; page 5, lines 121-122 and 129-130; page 6, line 150; page 10, lines 309-311).
- Page 8, Line 205: I do not understand the meaning of “Chao 1, Shannon, Simpson, observed species, and phylogenetic diversity indexes.”
Response: Thank you for this comment. These are some of the alpha diversity indices. Among the alpha diversity indices, those that showed a statistically significant increase are listed in parentheses.